# Health-Related Quality of Life and Related Factors in Persons with Preserved Ratio Impaired Spirometry: Data from the Korea National Health and Nutrition Examination Surve

**DOI:** 10.3390/medicina57010004

**Published:** 2020-12-23

**Authors:** I Re Heo, Ho Cheol Kim, Tae Hoon Kim

**Affiliations:** Department of Internal Medicine, Gyeongsang National University School of Medicine and Gyeongsang National University Changwon Hospital, Changwon 51472, Korea; h2hawk@naver.com (I.R.H.); hochkim@gnu.ac.kr (H.C.K.)

**Keywords:** preserved ratio impaired spirometry, chronic obstructive pulmonary disease, quality of life, Euro Quality of Life-5D, restrictive lung disease

## Abstract

*Background and Objectives*: preserved ratio impaired spirometry (PRISm) is a common spirometric pattern that causes respiratory symptoms, systemic inflammation, and mortality. However, its impact on health-related quality of life (HRQOL) and its associated factors remain unclear. We aimed to identify these HRQOL-related factors and investigate the differences in HROOL between persons with PRISm and those with normal lung function. *Materials and Methods:* we reviewed the Korea National Health and Nutrition Examination Survey data from 2008 to 2013 to evaluate the HRQOL of persons with PRISm, as measured while using the Euro Quality of Life-5D (EQ-5D) and identify any influencing factors. PRISm was defined as pre-bronchodilator forced expiratory volume in 1 s (FEV_1_) <80% predicted and FEV_1_ to forced vital capacity (FVC) ratio (FEV_1_/FVC) ≥0.7. Individuals with FEV_1_ ≥80% predicted and FEV1/FVC ≥0.7 were considered as Controls. *Results:* of the 27,824 participants over the age of 40 years, 1875 had PRISm. The age- and sex-adjusted EQ-5D index was lower in the PRISm group than in the control group (PRISm, 0.930; control, 0.941; *p* = 0.005). The participants with PRISm showed a significantly higher prevalence of hypertension (*p* < 0.001), diabetes (*p* < 0.001), obesity (*p* < 0.001), low physical activity (*p* = 0.001), ever-smoker (*p* < 0.001), and low income (*p* = 0.034) than those in the control group. In participants with PRISm, lower EQ-5D index scores were independently associated with old age (*p* = 0.002), low income (*p* < 0.001), low education level (*p* < 0.001), and no economic activity (*p* < 0.001). Three out of five EQ-5D dimensions (mobility, self-care, and usual activity) indicated a higher proportion of dissatisfied participants in the PRISm group than the control group. *Conclusions:* the participants with PRISm were identified to have poor HRQOL when compared to those without PRISm. Old age and low socioeconomic status play important roles in HRQOL deterioration in patients with PRISm. By analyzing risk factors that are associated with poor HRQOL, early detection and intervention of PRISm can be done in order to preserve patients’ quality of life.

## 1. Introduction

Preserved ratio impaired spirometry (PRISm) is a common spirometric pattern that causes respiratory symptoms, systemic inflammation, and mortality. Obstructive spirometric patterns, such as chronic obstructive pulmonary disease (COPD), have long been the main focus of study. The Global Initiative for Chronic Obstructive Lung Disease (GOLD) spirometrically defines COPD as a forced expiratory volume in 1 s (FEV_1_) to a forced vital capacity (FVC) ratio of less than 0.7 [1,2]. However, this definition overlooks patients with simultaneously reduced FEV_1_ and FVC [3,4]. Recently, these patients have been classified as having PRISm, being defined as post-bronchodilator FEV_1_ <80% of predicted and FEV_1_/FVC ≥0.7 [4,5,6,7]. Unlike patients with COPD, these patients have been mostly excluded from major clinical trials. Therefore, the manifestations, prognosis, and appropriate treatment of PRISm remain largely unknown.

FEV_1_ is the most commonly used marker for stratifying the severity of COPD. However, airway limitation does not fully reflect a patient’s symptoms, prognosis, and response to treatment [8]. On the other hand, health-related quality of life (HRQOL) incorporates the patient’s wellbeing, disease, and adaptation to treatment [9]. While the St. George’s Respiratory Questionnaire (SGRQ) has been applied as an important outcome measure in COPD [10,11], the deficits in the HRQOL of patients with PRISm are not well known.

Therefore, this study used nationally representative data from 2008 to 2013 from the Korea National Health and Nutrition Examination Survey (KNHANES) in order to identify factors related to HRQOL and investigate the differences in HRQOL between persons with PRISm and those with normal lung function.

## 2. Materials and Methods

### 2.1. Data Source and Participants

We used KNHANES data from 2008 to 2013. The KNHANES is a nationwide cross-sectional survey of health behaviors and nutrition status and a health examination study that was conducted by the Korean Centers for Disease Control and Prevention (KCDC) [12]. The KCDC conducts annual surveys by selecting a sample group that represents people over the age of one in South Korea. Detailed information on the survey plan and progress can be found on the KNHANES website (http://knhanes.cdc.go.kr).

Among the 27,824 participants ≥40 years of age, 1875 participants who had pre-bronchodilator FEV_1_ <80% of predicted and FEV_1_/FVC ≥0.7 were enrolled as the PRISm group. Another 14,467 participants who had FEV_1_ ≥80% of predicted and FEV1/FVC ≥0.7 were enrolled as the control group.

The research ethical review board of the KCDC approved the survey (2008-04EXP-01-C, 2009-01CON-03-2C, 2010-02CON-21-C, 2011-02CON-06-C, 2012-01EXP-01-2C, and 2013-07CON-03-4C), and all of the participants provided informed consent. The Institutional Review Board (GNUCH-2020-11-003) of Gyeongsang National University Changwon Hospital approved this study.

### 2.2. Variables

Pulmonary function tests were performed by trained paramedics using a rolling dry-seal spirometer (Vmax series 2130; SensorMedics Corp., Yorba Linda, CA, USA), according to the American Thoracic Society/European Respiratory Society guidelines [13]. HRQOL was measured while using the Euro Quality of Life-5D (EQ-5D) questionnaire, which consists of the following five dimensions: mobility, self-care, usual activity, pain/discomfort, and anxiety/depression [14]. The participants answered items in each dimension in one of three ways: no problem, moderate problem, or severe problem. The EQ-5D index was determined based on the participants’ responses to the five dimensions while using the reference value specific to the South Korean population [15].

Clinical and demographic data, such as age, sex, body mass index (BMI), hypertension, diabetes, hypercholesterolemia, smoking habits, alcohol consumption, and regular exercise, were collected. Socioeconomic variables, including income level, marital status, educational level, and level of economic activity, were reviewed. The participants were divided into two groups according to age (middle age: 40–59 and old age: ≥60 years) and four groups according to the Asia-Pacific BMI classification (underweight: <18.5, normal: 18.5–22.9, overweight: 23–24.9, and obese: ≥25.0) [16]. Ever-smokers were defined as those who had smoked more than 100 cigarettes (five packs) in their lifetimes. Binge-drinking was defined as more than five binge-drinking episodes (male: five drinks in 2 h; female: four drinks in 2 h) in the last month. Regular walking was defined as walking for 30 min. at least five times a week in the last month. Diabetes was defined as diagnosis by a physician, a fasting serum glucose level of ≥126 mg/dL, or having undergone diabetes treatment. Prediabetes was defined as a fasting serum glucose level between 100 mg/dL and <126 mg/dL. Hypertension was defined as diagnosis by a physician, having undergone treatment for hypertension, a systolic blood pressure of ≥140 mmHg, or a diastolic blood pressure of ≥90 mmHg. Normal blood pressure was defined as systolic blood pressure of <120 mmHg and a diastolic blood pressure of ≤80 mmHg. Hypercholesterolemia was defined as having undergone treatment for hypercholesterolemia or serum total cholesterol level of ≥240 mg/dL.

The socioeconomic variables were divided into two groups and analyzed. Household income level was divided into low income (lowest quartile) and middle-high income (medium-lowest, medium-highest, and highest quartile). For participants who were single, separated, or divorced, marital status was categorized as No. Educational level was classified as middle school graduation or lower (≤9 years) and high school education or higher. For participants who were unemployed or economically inactive, economic activity was categorized as No. We used the survey data, that was collected for investigating pulmonary tuberculosis, in order to check the participants’ respiratory symptoms, including cough, sputum, blood-tinged sputum, chest pain, dyspnea, fatigue, weight loss, and fever.

### 2.3. Statistical Analysis

We carried out a complex sample analysis while using stratification variables and weights. Socioeconomic and clinical demographic variables were compared between the PRISm and control groups. Continuous variables, which were presented as means (standard errors of the mean), were analyzed in two groups using the independent *t*-test. Categorical variables, which were presented as percentages (standard error of percentage), were compared using the chi-square test.

HRQOL, estimated using the EQ-5D index, was compared between the PRISm and control groups using a general linear regression model after adjustment for age and sex. A univariate analysis was performed in order to explore factors influencing HRQOL in persons with PRISm. A risk-adjusted analysis was then conducted to determine the independent factors affecting HRQOL. Variables with *p*-value <0.20 in the univariate study were included in the multivariate analysis. A *p*-value <0.05 was to be considered statistically significant. Statistical analysis was performed while using SPSS version 24.0 (IBM Corp., Armonk, NY, USA).

## 3. Results

### 3.1. Characteristics of Study Participants

Table 1 shows the socioeconomic status and clinical demographics of the patients. The mean ages of the PRISm and control groups were 54.5 ± 0.3 years and 53.7 ± 0.1 years, respectively. A total of 50.6% of the PRISm group and 57% of the control group were female. The parameters of lung function in participants with PRISm were as follows: mean FVC, 2.92 L (75.5% predicted); mean FEV_1_, 2.25 L (74.2% predicted); and, mean FEV1/FVC, 0.77. The PRISm group showed a significantly higher prevalence of hypertension (*p* < 0.001), diabetes (*p* < 0.001), obesity (*p* < 0.001), and physical inactivity (*p* = 0.001), when compared to the control group. The participants with PRISm had a significant history of smoking (*p* < 0.001). The proportion of participants with a low-income was higher in the PRISm group than in the control group (*p* = 0.034). There were no differences in marital status, educational level, and economic activity between the PRISm and control groups.

### 3.2. Comparing HRQOL between the PRISm and Normal Lung Function Groups

HRQOL was compared between the PRISm and control groups. In risk-adjusted analysis with age and sex, the mean EQ-5D index was significantly lower in participants with PRISm than those without PRISm (PRISm, 0.931 ± 0.004; control, 0.941 ± 0.001; *p* = 0.005). Among the participants with PRISm, 24.6% were “dissatisfied persons” (EQ-5D index < 0.9), as compared to only 20.5% “dissatisfied persons” among those in the control group (Figure 1).

More participants with PRISm experienced sputum symptoms (PRISm 10.9% vs. control 8.4%, *p* = 0.017), and significantly fewer experienced weight loss (PRISm 0.6% vs. control 1.5%, *p* = 0.039) than those in the control group. However, there were no differences in the percentages of participants that complained of other respiratory symptoms, including cough (PRISm 4.0% vs. control 4.8%, *p* = 0.253), blood-tinged sputum (PRISm 0.2% vs. control 0.1%, *p* = 0.525), chest pain (PRISm 2.0% vs. control 2.9%, *p* = 0.092), dyspnea (PRISm 1.2% vs. control 1.0%, *p* = 0.560), fatigue (PRISm 8.4% vs. control 7.7%, *p* = 0.499), and fever (PRISm 0.6% vs. control 0.8%, *p* = 0.534).

### 3.3. Factors Associated with HRQOL in PRISm

Table 2 shows the risk factors related to HRQOL deficits in the PRISm group.

In the univariate analysis, it was confirmed that old age (*p* < 0.001), female sex *(p* < 0.001), smoking habits (*p =* 0.025), binge-drinking (*p* < 0.001), diabetes (*p* = 0.006), hypercholesterolemia (*p* = 0.031), and poor socioeconomic factors (*p* < 0.001) affected HRQOL. In the risk-adjusted analysis with confounding factors, including socioeconomic status and clinical demographics, the EQ-5D index was found to be significantly lower in persons who were older (*p* = 0.002), earned a lower income (*p* < 0.001), had a lower educational level (*p* < 0.001), and were not economically active (*p* < 0.001) (Figure 2).

### 3.4. HRQOL According to Each Dimension of EQ-5D

Each dimension of the EQ-5D was categorized into moderate-to-severe or no problems. We investigated the distribution of people with problems across five dimensions (Figure 3). In terms of mobility, 18.3% of participants with PRISm complained of discomfort in mobility; in comparison, 15.9% of those in the control group complained of discomfort in mobility (*p* = 0.037). In the self-care dimension, 5.1% of participants with PRISm were not satisfied, while 3.6% of those in the control group were not satisfied (*p* = 0.009). In the usual activity dimension, 12.7% and 9.1% of participants complained of discomfort in daily activities in the PRISm and control groups, respectively (*p* < 0.001). However, there were no differences between the groups in the dissatisfaction rates of the pain/discomfort *(p* = 0.497) and anxiety/depression dimensions (*p* = 0.101).

## 4. Discussion

This study reviewed the KNHANES data from 2008 to 2013 in order to evaluate the HRQOL in persons with PRISm and identify any influencing factors. Overall, persons with PRISm had lower HRQOL than those without PRISm. Among people with PRISm, poor EQ-5D index scores were related to old age, low income, low educational level, and no economic activity. They complained of more phlegm and less weight loss than those in the control group did. Among the five dimensions of EQ-5D, persons with PRISm expressed a higher proportion of dissatisfaction in the dimensions of mobility, self-care, and usual activity, as compared to those with normal lung function.

PRISm is a relatively common spirometric pattern with a prevalence of between 3% and 20% [3,7,17,18,19,20]. However, most previous studies have been conducted on patients with restrictive lung disease (for example, patients with FEV_1_/FVC ≥0.70 and FVC <80%) [18,19,21,22]. This definition of restrictive lung disease is similar to that of PRISm, but it is not the same. Therefore, we focused on small groups that fit the definition of PRISm, and we applied the diagnostic criteria for PRISm that were endorsed by the COPDGene study (FEV_1_/FVC ≥ 0.70 and FEV_1_ < 80%) [6]. A recent study showed that patients with PRISm had a poorer HRQOL as compared to those with stage 0 COPD [23]. Our findings also showed that some patients with PRISm had impaired HRQOL. Previous studies showed that some respiratory disease, including COPD, asthma, and chronic cough, were associated with decreased HRQOL [24,25,26,27].

When compared to the control group, the PRISm group had a lower percentage of women, more people exposed to smoking, lower regular walking rates, and higher rates of hypertension, diabetes, and obesity. These results are consistent with the previously reported analyses of restrictive lung disease [28,29,30,31]. They suggested that physical inactivity, metabolic syndrome, and systemic inflammation were risk factors for, and markers of, FVC decline. Other studies have shown that patients with PRISm comprise a heterogeneous population [6,7,32], the subgroups of which have high cardiovascular burden and early mortality [7]. Therefore, we propose that PRISm may be a respiratory manifestation pattern in adult lifestyle disorders, and further research will be needed.

In terms of HRQOL decline, the patterns of PRISm are different from those of COPD. In a previous study of COPD, a decrease in HRQOL was observed, even at GOLD stage 0 [33]. It was shown that the total, impact, and symptom scores of SGRQ were increased, but not the activity score. Another study showed that chronic respiratory symptoms were more important markers for impairment of HRQOL than FEV_1_ in GOLD stages 0 and 1 [34]. However, the group with GOLD stage 4 had significantly decreased HRQOL [24]. In addition, a previous a study showed that asthma was associated with impaired HROOL, particularly in anxiety/depression dimension [27]. However, other studies of PRISm suggested that the HRQOL only deteriorated in the physical component and not the mental component of the 36-Item Short Form Health Survey (SF-36) [21]. This pattern is consistent with our findings, where the patients reported dissatisfaction in mobility, self-management, and daily activities. In the EQ-5D, these three dimensions reflect the physical component, when compared to the pain/discomfort or anxiety/depression dimensions. In addition, respiratory symptoms were not independently associated with impaired HRQOL in PRISm. However, further research on the role of respiratory symptoms is needed in symptomatic patients with PRISm who visited a hospital.

Our findings suggest that old age, low income, low educational level, and no economic activity may be markers of HRQOL deficits in PRISm. Similarly, a previous study showed that a restrictive spirometric pattern is associated with lower QOL, regardless of respiratory symptoms [21]. Our study also found that complaints of coughing or phlegm did not affect HRQOL. Moreover, obesity plays an important role in reducing FVC in patients with PRISm [20], and it is known to confer increased mortality [7,35]. However, being underweight, overweight, or obese was not associated with a deterioration of HRQOL in this study. Specifically, socioeconomic factors, rather than the patient’s clinical characteristics (except for the patients age), played an important role in the HRQOL of patients with PRISm. Because this study was conducted in a population-based setting in individuals with less severe symptoms, the socioeconomic factors exerted more pronounced effects on the changes in HRQOL than did the clinical characteristics. In patients with PRISm with low socioeconomic conditions, early intervention is needed in order to prevent the deterioration of HRQOL.

This study has several limitations. First, because this study was conducted while using data from a survey, it was difficult for the researchers to ensure the complete veracity of the responses. Second, because we used data from existing surveys, we could not apply a respiratory-specific questionnaire, such as the SGRQ, in order to study its impact on HRQOL. Third, the lung function test was conducted while using pre-bronchodilator spirometry. Fourth, although there was a statistically significant difference in the EQ-5D values between the PRISm and the control groups, this difference was small. This is presumed to be because the study participants were from the general population, and they were not patients. In studies that were conducted on other respiratory diseases, a wide range of minimal clinically important difference (MCID) values were observed [36,37]. The MCID in PRISm is not established, and further research is needed in order to clarify this. Finally, a cause-and-effect relationship could not be clarified, because of the cross-sectional nature of the KNHANES.

These limitations notwithstanding, the statistical reliability of the data from the KNHANES, a large-scale population-based survey that constitutes national data, which was used in this study, is guaranteed. Because the KNHANES represents the entire population of South Korea, selection bias has been minimized. Moreover, we confirmed not only unsatisfactory HRQOL in participants with PRISm, but also independent factors that are associated with HRQOL deficits in these patients.

## 5. Conclusions

PRISm is a relatively common spirometric pattern and it has been underdiagnosed in the general population. Participants with PRISm were identified to have poor HRQOL when compared to those without PRISm. In patients with PRISm, old age and low socioeconomic status were independent risk factors for the deterioration of HRQOL. Identifying the factors and mechanisms underlying PRISm may contribute to effective strategies for improving the HRQOL of these patients.

## Figures and Tables

**Figure 1 medicina-57-00004-f001:**
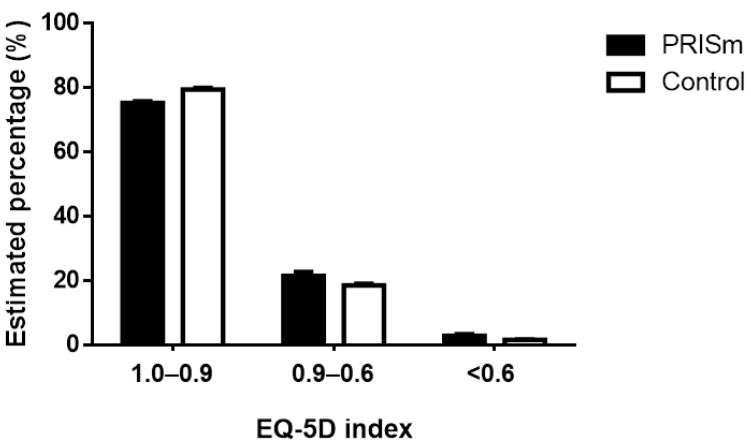
Distribution according to the degree of decline in health-related quality of life. Abbreviations: PRISm, preserved ratio impaired spirometry; EQ-5D, Euro Quality of Life-5D.

**Figure 2 medicina-57-00004-f002:**
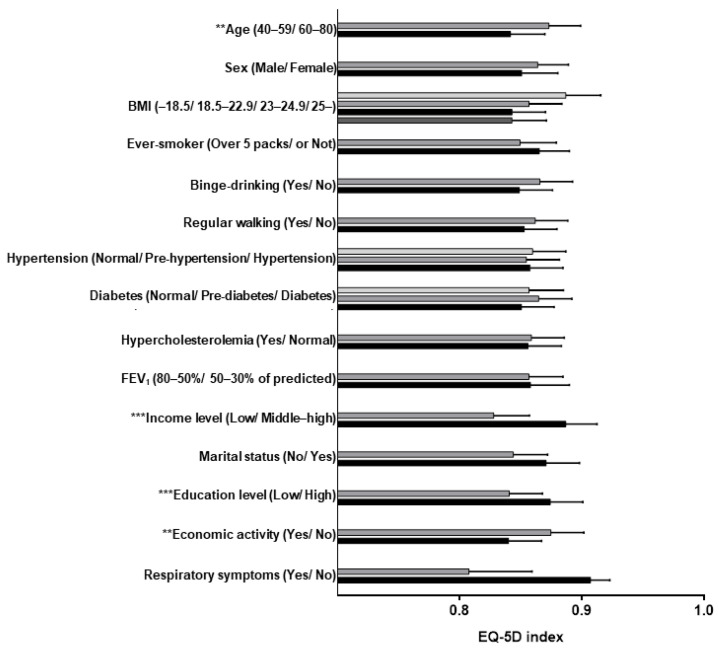
Factors influencing overall health-related quality of life in patients with preserved ratio impaired spirometry. Abbreviations: BMI, body mass index; FEV_1_, forced expiratory volume in 1 s. Note: ** *p* < 0.01, *** *p* < 0.001.

**Figure 3 medicina-57-00004-f003:**
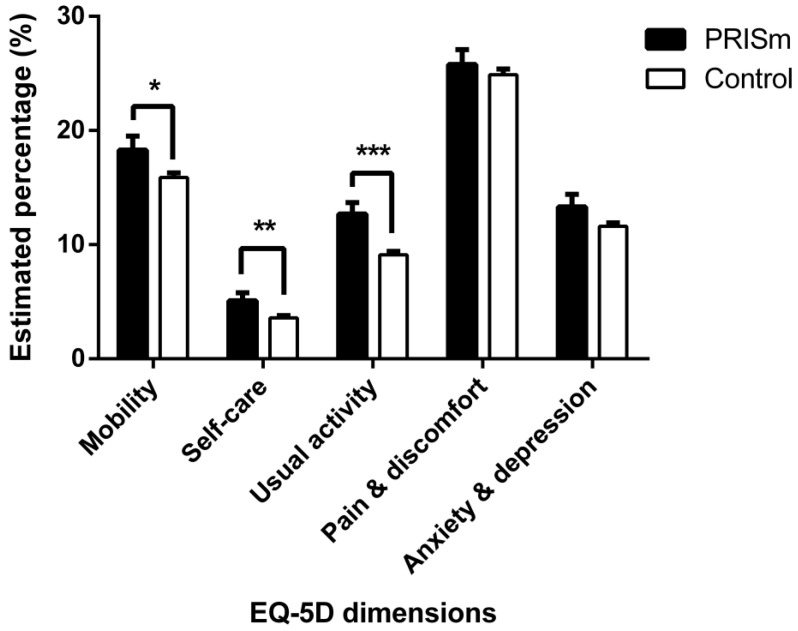
Dissatisfaction with health-related quality of life dimensions according to the Euro Quality of Life-5D (EQ-5D). Note: * *p* < 0.05, ** *p* < 0.01, *** *p* < 0.001.

**Table 1 medicina-57-00004-t001:** Baseline characteristics.

PRISm (*n* = 1875)		Control (*n* = 14,467)	*p*-Value
Age (years)			0.025 *
40–59	70.3 (1.4)	73.4 (0.6)	
60 and above	29.7 (1.4)	26.6 (0.6)	
Sex			<0.001 ***
Male	49.4 (1.6)	43.0 (0.5)	
Female	50.6 (1.6)	57.0 (0.5)	
BMI			<0.001 ***
<18.5	2.5 (0.5)	1.1 (0.1)	
18.5–22.9	27.6 (1.3)	33.8 (0.5)	
23–24.9	27.2 (1.4)	27.3 (0.5)	
>25.0	42.8 (1.5)	37.8 (0.6)	
Smoking habit			
Ever-smoker (>5 packs)	44.8 (1.6)	38.2 (0.5)	<0.001 ***
Current smoker	34.2 (1.5)	26.3 (0.5)	<0.001 ***
Binge-drinking	18.0 (1.3)	16.9 (0.4)	0.429
Regular walking exercise	35.0 (1.5)	39.1 (0.6)	0.001 **
Hypertension			<0.001 ***
Normal	30.3 (1.4)	36.9 (0.6)	
Pre-hypertension	25.6 (1.5)	27.5 (0.5)	
Hypertension	44.1 (1.6)	35.6 (0.6)	
Diabetes mellitus			<0.001 ***
Normal	55.9 (1.6)	65.0 (0.5)	
Pre-diabetes	27.7 (1.5)	24.4 (0.5)	
Diabetes mellitus	16.3 (1.1)	10.6 (0.3)	
Hypercholesterolemia	18.2 (1.2)	16.7 (0.4)	0.217
Lung function tests			
FVC (L)	2.92 (0.02)	3.63 (0.01)	<0.001 ***
FVC (% of predicted)	75.5 (0.3)	96.1 (0.1)	<0.001 ***
FEV_1_ (L)	2.25 (0.02)	2.89 (0.01)	<0.001 ***
FEV_1_ (% of predicted)	74.2 (0.2)	97.1 (0.1)	<0.001 ***
FEV_1_/FVC	0.77 (0.0)	0.80 (0.0)	<0.001 ***
Socioeconomic factors			
Low income	20.3 (1.3)	17.6 (0.5)	0.034 *
No marital status	15.3 (1.1)	16.0 (0.4)	0.521
Low educational level	34.6 (1.5)	35.7 (0.7)	0.479
No economic activity	35.6 (1.5)	33.9 (0.6)	0.306

Abbreviations: BMI, body mass index; FEV_1_, forced expiratory volume in 1 s; FVC, forced vital capacity; PRISm, preserved ratio impaired spirometry; FEV_1_, forced expiratory volume in 1 s. Note: data are presented as numbers (%) or mean ± standard deviation unless otherwise indicated. * *p* < 0.05, ** *p* < 0.01, *** *p* < 0.001.

**Table 2 medicina-57-00004-t002:** Factors associated with health-related quality of life in the preserved ratio impaired spirometry (PRISm) group.

Univariate Analysis			Risk-Adjusted Analysis	
Contrast Estimate (SE)		*p* Value	Contrast Estimate (SE)	*p* Value
Age (years)				
40–59	-	-	-	-
60 and above	−0.089 (0.010)	<0.001 ***	−0.031 (0.011)	0.004 **
Sex				
Male	-	-	-	-
Female	−0.039 (0.008)	<0.001 ***	−0.013 (0.014)	0.349
BMI				
<18.5	0.013 (0.015)	0.378	0.030 (0.016)	0.067
18.5–22.9	-	-	-	-
23–24.9	−0.017 (0.010)	0.095	−0.014 (0.009)	0.123
>25.0	−0.013 (0.009)	0.142	−0.014 (0.009)	0.124
Ever-smoker (>5 packs)	0.009 (0.004)	0.025 *	−0.016 (0.013)	0.222
Binge-drinking	0.041 (0.008)	<0.001 ***	0.017 (0.010)	0.091
Regular walking exercise	0.011 (0.008)	0.129	0.009 (0.006)	0.154
Hypertension				
Normal	-	-	-	-
Pre-hypertension	−0.016 (0.009)	0.083	−0.005 (0.009)	0.571
Hypertension	−0.013 (0.009)	<0.001 ***	−0.002 (0.009)	0.821
Diabetes mellitus				
Normal	-	-	-	-
Pre-diabetes	−0.010 (0.010)	0.32	0.008 (0.009)	0.376
Diabetes mellitus	−0.030 (0.011)	0.006 **	−0.006 (0.013)	0.635
Hypercholesterolemia	−0.024 (0.011)	0.031 *	0.003 (0.012)	0.806
Lung function tests				
FEV_1_ 50–80% of predicted	-	-	-	-
FEV_1_ 30–50% of predicted	0.004 (0.026)	0.864	-	-
Respiratory symptoms	−0.109 (0.065)	0.096	−0.099 (0.055)	0.072
Socioeconomic factors				
Low income	−0.102 (0.014)	<0.001 ***	−0.059 (0.015)	<0.001 ***
No marital status	−0.073 (0.014)	<0.001 ***	−0.027 (0.016)	0.093
Low educational level	−0.087 (0.009)	<0.001 ***	−0.034 (0.010)	<0.001 ***
No economic activity	−0.067 (0.009)	<0.001 ***	−0.035 (0.010)	<0.001 ***

Abbreviations: BMI, body mass index; FEV_1_, forced expiratory volume in 1 s; SE, standard error. Note: respiratory symptoms—report of cough or phlegm for one month. * *p* < 0.05, ** *p* < 0.01, *** *p* < 0.001.

## Data Availability

Publicly available datasets were analyzed in this study. This data can be found here: https://knhanes.cdc.go.kr/knhanes/main.do.

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
