# Peer review of "Health-Related Quality of Life and Related Factors in Persons with Preserved Ratio Impaired Spirometry: Data from the Korea National Health and Nutrition Examination Surve"

_medicina, 2020, doi:10.3390/medicina57010004_

Round 1

Reviewer 1 Report

This cohort study provides interesting data to support a hypothesis that PRISm is a lifestyle disorder linked perhaps to poor lung growth attendant on socioeconomic status, or other aspects of lifestyle and with impact on quality of life. The cohort is a substantial size, is adequately described and statistical analysis is appropriate. The English is clear and the manuscript flows and is easy to read. The discussion clearly shows how this paper might differ from others in the field. I had a few minor questions

  1. BMI was higher in PRISm patients - how common was obesity in the group? The ranges given at the moment only go up as far as 'overweight' (>25), where it is weights much higher that might usually be thought to predispose to co-morbidity observed in the group such as diabetes and hypertension
  2. A lot of statistical analyses were done - was correction for multiple testing applied and if not why not?
  3. Figure 2 - typo 'waling' instead of 'walking'

Author Response

Responses to Reviewer 1’s comments

This cohort study provides interesting data to support a hypothesis that PRISm is a lifestyle disorder linked perhaps to poor lung growth attendant on socioeconomic status, or other aspects of lifestyle and with impact on quality of life. The cohort is a substantial size, is adequately described and statistical analysis is appropriate. The English is clear and the manuscript flows and is easy to read. The discussion clearly shows how this paper might differ from others in the field. I had a few minor questions

Comment 1. BMI was higher in PRISm patients - how common was obesity in the group? The ranges given at the moment only go up as far as 'overweight' (>25), where it is weights much higher that might usually be thought to predispose to co-morbidity observed in the group such as diabetes and hypertension

→ Response: We are grateful to you for this question. As you know, the WHO defines overweight as a body mass index (BMI) of 25–29.9 Kg/m2 and obesity as a BMI of 30 Kg/m2 or more. However, in the Korea National Health and Nutrition Examination Survey (KNHANES), the Asia-Pacific BMI classification was applied to classify the individuals by weight. In Asia-Pacific countries, including Korea, overweight is defined as a BMI of 23.0­–24.9 Kg/m2 (Asia Pac J Clin Nutr. 2008;17(3):370–374). In addition, Asian individuals have a higher risk of developing comorbidities such as cardiovascular disease and type 2 diabetes mellitus at BMIs lower than 25 Kg/m2 (Lancet. 2004;363(9403):157–163). Thus, in this study, we considered a BMI of 25 Kg/m2 as the cut-off point for obesity. Furthermore, 6.0% of the PRISm (preserved ratio impaired spirometry) group and 3.7% of the control group had a BMI of 30 Kg/m2 or higher.

We have revised the manuscript as per your comment (page 2, line 87).

Comment 2. A lot of statistical analyses were done - was correction for multiple testing applied and if not why not?

→ Response: The KNHANES has been conducted with an annual rolling sampling design, including a complex, stratified, multistage probability-cluster survey of a representative Korean population sample aged ≥ 1 year. We used stratification variables and sampling weights in all analyses in consideration of the complex sampling design of KNHANES.

In the analysis of risk factors related to health-related quality of life (HRQOL) deficit in the PRISm group, all the variables from the univariate analysis were used as variables in the multivariate analysis. To solid the results, the variables considered significant (p < 0.2 in univariate analysis) were then included in further multivariate analysis. However, the results were the same even when the multivariate analyses were performed by only using variables with p-values of <0.20 in univariate analysis.

We have revised the manuscript as per your comment (page 3, line 113; page 8, Table 2).

Comment 3. Figure 2 - typo 'waling' instead of 'walking'

→ Response: We apologize for the inconvenience.

We have revised Figure 2 as per your comment (page 10, Figure 2).

Reviewer 2 Report

Thank you for the opportunity to review this manuscript, in which the authors focus on health-related quality of life(HRQL) in persons with preserved ratio impaired spirometry (PRISm), and associations of HRQL with other risk factors. they show that HRQL is impaired in PRISm compared to persons with normal spirometry, especially impacting mobility and usual activities, and report associations with cardiovascular risk factors suggesting that PRISm is a respiratory manifestation of individuals with high cardiovascular risk burden.   I do have a series of comments that I would like the authors to address.
  1. while a comparison of individuals with PRISm vs. individuals with preserved/normal lung function is helpful in showing the impact of PRISm, it only gives a very limited assessment of HRQL in PRISm. In the discussion section, the authors cite others who have compared HRQL in PRISm with (early stage) COPD. I was therefore wondering if the authors had considered comparing HRQL of individuals with PRISm with individuals with COPD, asthma, or other (chronic) conditions in the KNHANES dataset. I feel that such a comparison would really strengthen the quality of the manuscript, by being able to gauge in the same population the impact of PRISm vs. other conditions
  2. discussion of the HRQL in PRISm would additionally benefit from a focus on effect sizes (e.g. what are meaningful effect sizes in HRQL) or benchmarking with other conditions to further qualify the HRQL differences that have been observed in PRISm vs controls.
  3. when discussion risk factors, the authors discuss differences in risk factors for individuals with PRISm vs. controls, and association of risk factors with HRQL within the PRISm group. In the latter analysis, the risk factors they identify (old age, low income, low education, and no economic activity) may not be unique to PRISM but also contribute to HRQL in the control group. Would the authors be able to report these analyses for the control group and investigate whether the magnitude of these associations is different for both groups?
  4. In terms of symptom burden, the authors report differences in sputum, but not other respiratory symptoms for PRISm vs control individuals; they also report on weight-loss differences. Could the authors provide more information on the total set of symptoms that were queried, and the selections they made (i.e. why include weight loss, which respiratory symptoms to include, why not include other symptoms).
  5. for continuous data that were dichotomized or categorized (e.g. age, ever smokers, binge drinking, income), please report the rationale (why did the authors choose this grouping), in the methods section, or provide sensitivity analyses for other groupings.
  6. for risk adjusted analyses, please consider describing the selection for including variables in the adjusted models and/or providing sensitivity analyses for other models. in the current version of the paper, it is not always clear if and why included variables differ in different models.

Author Response

Responses to Reviewer 2’s comments

Thank you for the opportunity to review this manuscript, in which the authors focus on health-related quality of life(HRQL) in persons with preserved ratio impaired spirometry (PRISm), and associations of HRQL with other risk factors. they show that HRQL is impaired in PRISm compared to persons with normal spirometry, especially impacting mobility and usual activities, and report associations with cardiovascular risk factors suggesting that PRISm is a respiratory manifestation of individuals with high cardiovascular risk burden. I do have a series of comments that I would like the authors to address.

Comment 1. while a comparison of individuals with PRISm vs. individuals with preserved/normal lung function is helpful in showing the impact of PRISm, it only gives a very limited assessment of HRQL in PRISm. In the discussion section, the authors cite others who have compared HRQL in PRISm with (early stage) COPD. I was therefore wondering if the authors had considered comparing HRQL of individuals with PRISm with individuals with COPD, asthma, or other (chronic) conditions in the KNHANES dataset. I feel that such a comparison would really strengthen the quality of the manuscript, by being able to gauge in the same population the impact of PRISm vs. other conditions

→ Response: We appreciate your valuable comment. Like you pointed out, we also wanted to use the KNHANES data to study whether patients with COPD, asthma, and chronic cough had a lower HRQOL than healthy individuals. However, through preliminary research, we confirmed that a significant portion of the studies cited by us (the studies you refer to in your comment) had been conducted using the KNHANES data. A previous study showed that asthma was associated with impaired HRQOL, particularly when it co-occurred with other severe conditions such as anxiety/depression (J Asthma. 2018 Sep;55(9):1011-1017). Another study showed that the higher the COPD severity, the lower the quality of life (Int J Chron Obstruct Pulmon Dis. 2016 Jan 13;11:103-9). A third study reported significant associations between chronic cough and HRQOL in adults aged ≥40 years (Allergy Asthma Immunol Res. 2020 Nov;12(6):964-979). In this study, we tried to discover the factors associated with HRQOL decline in patients with PRISm; this condition is commonly detected through lung function tests, but these patients are not diagnosed with COPD and are only observed. Furthermore, we not only confirmed the impairment of HRQOL in participants with PRISm, but also discovered independent factors associated with it.

We have revised the manuscript as per your comment (page 11. line 190; page 13. line 203).

Comment 2. discussion of the HRQL in PRISm would additionally benefit from a focus on effect sizes (e.g. what are meaningful effect sizes in HRQL) or benchmarking with other conditions to further qualify the HRQL differences that have been observed in PRISm vs controls.

→ Response: This study was conducted in the general population, not in patients who presented to a hospital with symptoms. Therefore, there was only a small decrease in HRQOL in both the control and PRISm groups. In other studies, conducted on patients, the MCID (minimal clinically important difference) values of the Euro Quality of Life-5D (EQ-5D) index were as follows: In COPD, the estimated pooled MCID for the EQ-5D index was 0.028 (BMC Pulm Med. 2020 Mar 23;20(1):73). Additionally, there were statistically significant differences in the MCIDs between respiratory patients with and without cardiovascular disease (-0.09), musculoskeletal disease (-0.14), kidney disease (-0.13), and endocrine disease (-0.13) (J Clin Med. 2019 Feb 7;8(2):214).

Although the HRQOL expressed by the EQ-5D index showed a statistically significant between-groups difference in this study, further research is needed to determine whether it has actual clinical significance. We have added this to the Discussion section of the manuscript as per your comment (page 13, line 225).

Comment 3. when discussion risk factors, the authors discuss differences in risk factors for individuals with PRISm vs. controls, and association of risk factors with HRQL within the PRISm group. In the latter analysis, the risk factors they identify (old age, low income, low education, and no economic activity) may not be unique to PRISM but also contribute to HRQL in the control group. Would the authors be able to report these analyses for the control group and investigate whether the magnitude of these associations is different for both groups?

→ Response: We analyzed the factors affecting HRQOL in the control group using the same methods that were used for the PRISm group. In the univariate analysis, lower EQ-5D index scores were associated with old age, female sex, obesity, never-smoker, no-binge-drinking, no-regular walking exercise, hypertension, diabetes, hypercholesterolemia, low income, no marital status, low education, and no economic activity. In the multivariate analysis, lower EQ-5D index scores were independently associated with old age, female sex, obesity, no-regular walking exercise, pre-hypertension, hypercholesterolemia, low income, no marital status, low education, and no economic activity. The relationship between HRQOL and some factors (old age, low income, low education, and no economic activity) showed similar patterns in both the PRISm and control groups. However, we confirmed that each group had different independent associated factors.

Table. Factors associated with health-related quality of life in the control group

Univariate analysis

Risk-adjusted analysis

Contrast estimate (SE)

p value

Contrast estimate (SE)

p value

Age (years)

40-59

-

-

-

-

60 and above

-0.082 (0.003)

<0.001***

-0.034 (0.004)

<0.001***

Sex

Male

-

-

-

-

Female

-0.045 (0.002)

<0.001***

-0.019 (0.004)

<0.001***

BMI (Kg/m2)

<18.5

0.006 (0.007)

0.411

0.015 (0.008)

0.057

18.5–22.9

-

-

-

-

23–24.9

-0.002 (0.003)

0.518

-0.002 (0.003)

0.398

>25.0

-0.012 (0.003)

<0.001***

-0.008 (0.003)

0.006**

Ever-smoker (>5 packs)

0.029 (0.002)

<0.001***

-0.005 (0.004)

0.158

Binge-drinking

0.034 (0.003)

<0.001***

0.004 (0.003)

0.168

Regular walking exercise

0.014 (0.002)

<0.001***

0.013 (0.002)

<0.001***

Hypertension

Normal

-

-

-

-

Pre-hypertension

-0.004 (0.003)

0.092

-0.006 (0.002)

0.014*

Hypertension

-0.033 (0.003)

<0.001***

-0.004 (0.003)

0.232

Diabetes mellitus

Normal

-

-

-

-

Pre-diabetes

-0.002 (0.003)

0.440

0.003 (0.003)

0.329

Diabetes mellitus

-0.034 (0.005)

<0.001***

-0.005 (0.004)

0.241

Hypercholesterolemia

-0.027 (0.003)

<0.001***

-0.008 (0.003)

0.017*

Respiratory symptoms

-0.018 (0.011)

0.100

-0.016 (0.011)

0.129

Socioeconomic factors

Low income

-0.087 (0.004)

<0.001***

-0.044 (0.005)

<0.001***

No marital status

-0.060 (0.004)

<0.001***

-0.019 (0.004)

<0.001***

Low educational level

-0.077 (0.003)

<0.001***

-0.036 (0.003)

<0.001***

No economic activity

-0.056 (0.003)

<0.001***

-0.021 (0.003)

<0.001***

Abbreviations: BMI, body mass index; FEV1, forced expiratory volume in 1 s; SE, standard error

Note: respiratory symptoms – report of cough or phlegm for 1 month

*p < 0.05, **p < 0.01, ***p < 0.001

Comment 4. In terms of symptom burden, the authors report differences in sputum, but not other respiratory symptoms for PRISm vs control individuals; they also report on weight-loss differences. Could the authors provide more information on the total set of symptoms that were queried, and the selections they made (i.e. why include weight loss, which respiratory symptoms to include, why not include other symptoms).

→ Response: The KNHANES was conducted on a group that reflects the entire population of South Korea. Questions about respiratory symptoms were included in the survey to check for pulmonary tuberculosis. In this study, since we were focusing not on tuberculosis screening, but on PRISm, we felt that there was not much meaning in assessing the respiratory symptoms. In addition, as the number of participants experiencing respiratory symptoms was small, it could not play a significant role in the analysis of the related factors. Nevertheless, there have been reports in a paper (Thorax. 2008 Sep;63(9):768-74) that the chronic response symptoms (chronic cough, sputum, and dyspnea on exertion) play an important role in HRQOL, so we applied the modified respiratory symptoms to multivariate analysis. Respiratory symptoms may have additional meaning when conducting research on patients who visited the hospital. Further research on respiratory symptoms is needed in symptomatic patients with PRISm who visited a hospital.

We have revised the manuscript as per your comment (page 4, line 103; page 8, lines 142–144; page 13, line 208).

Comment 5. for continuous data that were dichotomized or categorized (e.g. age, ever smokers, binge drinking, income), please report the rationale (why did the authors choose this grouping), in the methods section, or provide sensitivity analyses for other groupings.

→ Response: We have categorized our data by modifying the techniques used in previous studies. In this study, participants aged between 40 and 80 were enrolled. They were then classified into two groups according to age (middle age: 40–59 years and old age: over 60 years). Household income levels were classified based on those in the following studies: PLoS One. 2018 Apr 10;13(4):e0195713 and BMC Geriatr. 2020 Nov 26;20(1):509.

Ever-smoker was defined as a person who has smoked more than 100 cigarettes (5 packs) in their lifetime. This definition is based on the question "How many cigarettes have you smoked in your lifetime? 1) Less than 5 packs (100 pieces), 2) More than 5 packs (100 pieces), 3) never" investigating smoking history in the KNHANES. In addition, the National Health Interview survey defines smoking habits based on a cut-off of 100 cigarettes (https://www.cdc.gov/nchs/nhis/tobacco/tobacco_glossary.htm). The Centers for Disease Control and Prevention has defined binge-drinking as a pattern of drinking that brings a person’s blood alcohol concentration to 0.08 g/dl or more. This typically happens when men consume 5 or more drinks or women consume 4 or more drinks in about 2 hours (https://www.cdc.gov/alcohol/fact-sheets/binge-drinking.htm).

We have revised the manuscript as per your comment (pages 3–4, lines 83–103).

Comment 6. for risk adjusted analyses, please consider describing the selection for including variables in the adjusted models and/or providing sensitivity analyses for other models. in the current version of the paper, it is not always clear if and why included variables differ in different models.

→ Response: In this study, all the variables from the univariate analysis were included as variables in the multivariate analysis. To solid the results, the variables considered significant (p < 0.2 in univariate analysis) were then included in further multivariate analysis. However, the results were the same even when multivariate analyses were performed only using variables with p-values of <0.20 in the univariate studies. The Method section of the manuscript and Table 2 have been revised accordingly (page 3, line 113; page 8, Table 2).

Round 2

Reviewer 2 Report

Thank you for your response to the previous set of comments.

I have one further comment, which relates to the abstract:

the concluding paragraph of the abstract is not in correspondence with the concluding paragraph of the manuscript: The abstract is far less restrained compared to the manuscript, and therefore does not reflect the content of the manuscript well. Please consider changing the conclusion of the abstract.

Author Response

Responses to Reviewer 2’s comments

the concluding paragraph of the abstract is not in correspondence with the concluding paragraph of the manuscript: The abstract is far less restrained compared to the manuscript, and therefore does not reflect the content of the manuscript well. Please consider changing the conclusion of the abstract.

→ Response: We appreciate your valuable comment. We have added a sentence in the conclusion of the abstract to better reflect the content of the manuscript.

We have revised the manuscript as per your comment (page 1, line 29 – green-colored highlighted).